# Characterization, Diversity, and Structure-Activity Relationship Study of Lipoamino Acids from *Pantoea* sp. and Synthetic Analogues

**DOI:** 10.3390/ijms20051083

**Published:** 2019-03-02

**Authors:** Seindé Touré, Sandy Desrat, Léonie Pellissier, Pierre-Marie Allard, Jean-Luc Wolfender, Isabelle Dusfour, Didier Stien, Véronique Eparvier

**Affiliations:** 1CNRS, Institut de Chimie des Substances Naturelles, UPR 2301, Université Paris-Saclay, 1 avenue de la Terrasse, 91198 Gif-sur-Yvette, France; seindet@gmail.com (S.T.); sandy.desrat@cnrs.fr (S.D.); 2School of Pharmaceutical Sciences, EPGL, University of Geneva, University of Lausanne, Rue Michel Servet 1, CH-1211 Geneva 4, Switzerland; leonie.pellissier@unige.ch (L.P.); pierre-marie.allard@unige.ch (P.-M.A.); jean-luc.wolfender@unige.ch (J.-L.W.); 3Institut Pasteur de la Guyane (IPG), Unité de Contrôle et Adaptation des Vecteurs, 97306 Cayenne, Guyane, France; isabelle.dusfour@pasteur.fr; 4Sorbonne Université, CNRS, Laboratoire de Biodiversité et Biotechnologie Microbienne, LBBM, Observatoire Océanologique, 66650 Banyuls-sur-mer, France; didier.stien@cnrs.fr

**Keywords:** hydroxyacyl-phenylalanine, lipoamino acid, entomopathogen, bacteria, antimicrobial, larvicidal, *Aedes aegypti*

## Abstract

A biological evaluation of a library of extracts from entomopathogen strains showed that *Pantoea* sp. extract has significant antimicrobial and insecticidal activities. Three hydroxyacyl-phenylalanine derivatives were isolated from this strain. Their structures were elucidated by a comprehensive analysis of their NMR and MS spectroscopic data. The antimicrobial and insecticidal potencies of these compounds were evaluated, and compound **3** showed 67% mortality against *Aedes aegypti* larvae at a concentration of 100 ppm, and a minimum inhibitory concentration (MIC) of 16 µg/mL against methicillin-resistant *Staphylococcus aureus*. Subsequently, hydroxyacyl-phenylalanine analogues were synthesized to better understand the structure-activity relationships within this class of compounds. Bioassays highlighted the antimicrobial potential of analogues containing saturated medium-chain fatty acids (12 or 14 carbons), whereas an unsaturated long-chain fatty acid (16 carbons) imparted larvicidal activity. Finally, using a molecular networking-based approach, several close analogues of the isolated and newly synthesized lipoamino acids were discovered in the *Pantoea* sp. extract.

## 1. Introduction

The development of novel antimicrobial and insecticidal agents is urgently needed to improve public health worldwide. Indeed, drug-resistant pathogenic microorganisms such as methicillin-resistant *Staphylococcus aureus* (MRSA), vancomycin-resistant *S. aureus* (VRSA), and vancomycin-resistant *Enterococci* (VRE) have emerged [1]. Other Gram-negative pathogens are particularly worrisome because they are becoming resistant to nearly all of the antibiotic drugs currently available, including carbapenems [2]. These pathogens have become a major clinical problem, causing significant mortality in both healthy hosts and in those with underlying comorbidities [3]. Thus, it is essential to investigate new drugs to address the decreasing efficiency of the currently available antibiotics [2]. Another major public health problem concerns mosquitoes and their ability to spread pathogens such as malaria parasites, dengue, chikungunya, and more recently the Zika virus. The World Health Organization (WHO) estimated that more than 80% of the world’s population is at risk of contracting vector-borne diseases, and each year more than 700,000 people die as a result [4]. *Aedes* sp. mosquitoes are the primary vector for transmitting arboviruses worldwide [5]. Today, the two main approaches to control them involve genetic modification or the application of chemicals. Synthetic chemicals including pyrethroids or organophosphates such as temephos and malathion have been sprayed into the environment for decades. However, most of these synthetic chemicals have adverse effects, leading to the development of resistance, environmental pollution, and the introduction of toxic hazards to humans and other nontarget organisms [6]. Furthermore, the abuse and/or misuse of these compounds has resulted in the loss of vector control efficacy. Thus, specific safer pesticides are urgently needed [7]. 

Natural products have proven to be an immeasurable source of bioactive compounds [8]. Entomopathogenic microorganisms, the natural enemies of insects, are known to produce bioactive metabolites that have been implicated in complex defense and self-protection mechanisms. Indeed, to achieve ecological success, they produce several chemical entities including insecticides and/or antimicrobial metabolites [9]. 

As part of our investigation into the secondary metabolites produced by entomopathogenic microorganisms, a collection of 53 strains was extracted and screened against *Aedes aegypti* mosquito larvae and human pathogenic microorganisms (*Staphylococcus aureus*, MRSA, *Candida albicans*, and *Trichophyton rubrum*). The extract from the bacterium *Pantoea* sp. SNB-VECD14B exhibited a mortality rate of 97.2% against *Ae. aegypti*. larvae at a concentration of 100 ppm and a minimum inhibitory concentration (MIC) of 16 µg/mL against *T. rubrum*. In addition, the full strain collection was profiled by high-resolution mass spectrometry and the resulting fragmentation data were organized as a single molecular network. This molecular networking approach allowed us to organize untargeted tandem MS datasets according to their spectral similarity and thus to group analytes by structural similarity [10]. Similar approaches have served as powerful tools to navigate the chemical space of complex biological matrices and can be used to view the chemical constituents of a wide variety of extracts in a single map [11,12]. Appropriate taxonomical color mapping allowed us to highlight a structurally related series of compounds also produced by *Pantoea* sp. that was selected for further investigation.

## 2. Results and Discussion

### 2.1. Isolation and Structure Elucidation

The EtOAc extract of *Pantoea* sp. SNB-VECD14B was subjected to bioguided preparative high performance liquid chromatography (HPLC) using a C_18_ silica gel column to yield three pure compounds (**1**–**3**) (Figure 1). 

The molecular formula of compound **1** was determined to be C_23_H_37_NO_4_ via HRESI-TOFMS analysis which gave a pseudomolecular ion at *m*/*z* 392.2769 [M+H]^+^ (calculated for C_23_H_38_NO_4_^+^, 392.2795), indicating six degrees of unsaturation. The structure of this compound was deduced from NMR spectral data (Table 1). The ^1^H and ^13^C NMR spectra of **1** suggested the presence of one methyl group (δ_H_ 0.90) that appeared as a triplet in the ^1^H NMR spectrum, fatty acid methylenes integrating for 20 protons (δ_C_ 30.9, δ_H_ 1.29) that appeared as a characteristic broad signal in the ^1^H NMR spectrum, two other distinct methylenes at δ_H_ 1.40 and 2.29/2.34, and six methines, one of which was hydroxylated (δ_H_ 3.85) with the other five methines corresponding to aromatic protons with chemical shifts in the ^1^H NMR spectrum between 7.18 and 7.24 ppm. The sequence of ^1^H-^1^H COSY signals from H2′ to H5′ and from H12′ to H14′ allowed us to determine the presence of a hydroxylated fatty acid moiety. ^1^H-^13^C correlations observed in the HMBC spectrum between H2′/C1′, C3′ and C4′ confirmed that compound **1** has a tetradecanoate moiety that is hydroxylated at C3′ (Figure 2). Another ^1^H-^1^H correlation was observed between H2 and H3 and ^1^H-^13^C correlations between H2/C1, C3 and H3/C1, C2, C4, and C5 along with the C1 chemical shift at δ 176.0 demonstrated the presence of a phenylalanine unit. Although no correlation was observed (COSY, HMBC, NOESY) between the phenylalanine and fatty acid moieties, the only possible linkage between these two units is via an amide bond. This is consistent with the chemical shifts at C1 (δ_C_ 173.8), C2 (δ_C_ 55.9), and H2 (δ_H_ 4.60). Thus, compound **1** was identified as *N*-(3-hydroxytetradecanoyl)phenylalanine. At this stage, the asymmetric carbons configurations could not be determined. The stereochemistry of this molecule and the following compounds will be demonstrated by synthesis.

The molecular formula of compound **2** was determined to be C_25_H_39_NO_4_ based on the ion observed at *m/z* 418.2926 [M+H]^+^ in the HRESI-TOFMS experiment (calculated for C_25_H_40_NO_4_^+^_,_ 418.2952), which corresponded to seven degrees of unsaturation. The ^1^H and ^13^C NMR spectra revealed several similarities to **1**. Only minor differences were observed, including the presence of a broad methylene signal integrating for just fourteen protons, along with two de-shielded protons at δ_H_ 5.35 and two methylenes at δ_H_ 2.04 corresponding to the methines and adjacent methylenes of a double bond. The COSY and HMBC data of **2** allowed us to determine that the fatty acid moiety is 3-hydroxyhexadec-9-enoate and that it is hydroxylated at C3. The (*Z*) configuration of the double bond was determined by comparison with the literature data based on the distinctive splitting of the olefinic protons [13]. The smaller coupling constant in the (*Z*) double bond impacts on the spacing of the olefinic protons multiplet peaks and the overall aspect of the multiplet in ^1^H NMR. The double bond configuration was confirmed by comparison with analytical data of synthetic compounds described below. Thus, compound **2** was finally identified as (*Z*)-*N*-(3-hydroxyhexadec-9-enoyl)phenylalanine.

The HRESI-TOFMS experiment for compound **3** indicated a molecular formula of C_25_H_39_NO_3_ with a *m*/*z* of 402.3004 [M+H]^+^ (calculated for C_25_H_40_NO_3_^+^, 402.3003), which corresponded to seven degrees of unsaturation. After examining the ^1^H and ^13^C NMR spectra, compound **3** was obviously similar to **2**. The only differences observed include signals pertaining to the hydroxy group that are absent in compound **3** as well as differences in the signals of the neighboring protons and carbons (Table 1). Ultimately, the fatty acid moiety was determined to be hexadec-9-enoate. The COSY and HMBC spectra showed the same correlations as in **2**, along with a correlation between H2/C1′ that further confirmed the amide linkage between the two moieties. Compound **3** was therefore identified as (*Z*)-*N*-hexadec-9-enoylphenylalanine (Figure 2). The configuration of the amino acid subunit was determined to be L (*S*) after synthesis of **3** and ***ent*-3** (described below) and comparison of their optical rotations. All synthetic (2*S*)-lipoamino acids had a positive optical rotation, as did compounds **1*–*3**. Altogether, the analytical data along with the obvious biosynthetic resemblance between **1**, **2**, and **3** led to the conclusion that the absolute stereochemistry of the phenylalanine moiety was L in **1** and **2** as well. The configuration of C-3′ could not be determined. These three compounds were isolated for the first time from natural sources and the complete NMR data for molecules **1*–*3** is described for the first time in the literature [14,15]. 

### 2.2. Synthesis of Lipoaminoacid Analogues 

Our structure activity relationship (SAR) investigation began with the synthesis of various acylated phenylalanine analogues of compound **3**. We modified the carbon chain length of the fatty acid, the configuration of the amino acid (L vs. D), and the number of double bonds in the fatty acid alkyl chain (Scheme 1). The synthetic method used classical conditions for peptides coupling [16]. Amide coupling between l- or d- amino acid methyl esters and fatty acids in the presence of HATU and Hunig’s base, followed by saponification of the ester moiety with lithium hydroxide gave a series of 32 acylated phenylalanine derivatives. Thus, starting from the methyl ester of either l- or d-phenylalanine, 16 ester derivatives (compounds **4** to **19**) and 16 free acids (compounds **20** to **35**) were prepared with varying lengths of the fatty acid chain (from 12 to 20 carbons) and different degrees of unsaturation. Compounds **3**, **3-OMe**, ***ent*-3**, ***ent*-3-OMe**, **8**, **15**, **24**, and **31** were *Z*-alkenes, whereas compounds **9**, **16**, **25**, and **32** had an *E*-alkene configuration. Four additional lipoamino acids (**18**, **19**, **34**, **35**) were prepared from palmitoleic acid and the methyl ester of either l-alanine or l-tyrosine following the same synthetic procedures. The general synthetic process is shown in Scheme 1. The structures of the newly synthesized compounds were confirmed by NMR and HRESI-TOFMS analysis.

### 2.3. Biological Activities and SAR Investigation 

Compounds **1**–**3**, isolated from the bacteria *Pantoea* sp., were assayed for their antimicrobial and insecticidal activities against human pathogenic microorganisms and *Ae. aegypti* larvae (Table 2). The bioassays showed that compound **1** exhibited minimal larvicidal and antibacterial activities. In contrast, compound **3** demonstrated significant larvicidal activity with a mortality rate of 67.3% and antibacterial activity with an MIC of 16 µg/mL against methicillin-resistant *S. aureus*. Based on the observed differences of these compounds in the bioassays, we evaluated analogues of **1*–*3** to gain more insight into the structure-activity relationship (SAR) of this class of compounds.

The synthetic compounds were evaluated for their larvicidal and antimicrobial activities (Table 2). First, the results showed that the best antimicrobial activities were obtained using compounds **1**, **3**, **7**, **19**, **22**, **23**, **29**, **30**, **34**, and **35** (MIC ≤64 µg/mL against MRSA and/or *T. rubrum*) with the highest antimicrobial activity being observed with compounds **30** (MIC <8 µg/mL against MRSA and *T. rubrum*) and **7** (MIC = 8 µg/mL against MRSA). Compounds **19** and **23** only showed activity against *T. rubrum*, whereas compounds **1**, **3**, and **7** only showed antibacterial activity against MRSA. Finally, compound **29** was the only compound to demonstrate MICs ≤64 µg/mL on all tested pathogenic strains. Interestingly, compounds **1**, **3**, **7**, **23**, and **35** showed antibacterial activity against MRSA only and not *S. aureus*. Most of these compounds contain a carboxyl group and a saturated alkyl chain with 12 to 14 carbons. Similar biological activities were observed irrespective of the stereochemical configuration of the free acids (**22**–**35**). On the other hand, given the antibacterial activities of compounds **8** and **16** against MRSA, the D configuration appears deleterious for lipoaminoesters. 

Regarding larvicide potential, compounds **1**, **3**, ***ent*-3**, **19**, **23**, **24**, **27**, **31**, and **32** were the most active with mortality rates ranging from 26.5% to 71.1% at a concentration of 10 ppm. These compounds all possess long-chain acyl groups (mostly C16 and C18). Moreover, the compounds prepared from D-phenylalanine (***ent*-3**, **27**, **31**, and **32**) were more active than their L-phenylalanine counterparts (**3**, **20**, **24**, and **25**). We next evaluated the effect of alkyl chain unsaturation on larvicide potential and found that unsaturated molecules were more active than saturated ones. For example, the mortality rate of compound **27** was 23.3%, while it was 71.1% for ***ent*-3**. Likewise, the mortality rate of compound **33** was 2.3% while it was 53.3% for **31**. Finally, unsaturated compounds with the *Z-*configuration were found to be more active than those with the *E-*configuration (i.e., compounds **24** and **31** compared to **25** and **32**). 

We also investigated the effect of amino acid replacement on compound bioactivity. In general, the methyl esters of tyrosine alanine derivatives (**18** and **19**, respectively) were not antimicrobial. In contrast, the corresponding acids (**34** and **35**) appeared to be more active than the phenylalanine analogue **3**. The ester compounds were less soluble than the acids, which might explain their weaker activity. Regarding larvicidal activity, only the methyl ester of the acylated tyrosine derivative (**19**) was active (mortality rate of approximately 30%).

In summary, compound **3** was the only compound that demonstrated larvicidal and slight antibacterial activity with mortality rates of 42.9% and 67.3% and MIC values of 16 and 64 µg/mL against *Ae. aegypti* larvae and MRSA, respectively. 

### 2.4. Molecular Networking

A molecular network (MN) was generated to organize the tandem MS data acquired from the entire collection of entomopathogenic strains that were biologically tested in this work, including *Pantoea* sp. SNB-VECD14B. After applying appropriate data treatment to align the common features among the collection, the resulting spectral data were submitted to the Global Natural Products Social molecular networking (GNPS) platform for molecular network generation [10,11,12]. The resulting molecular network grouped 20,859 spectra into 2619 clusters (Figure 3). Taxonomical mapping was applied by attributing a given color code to each strain in the collection, allowing us to visually navigate through the MN and check for the presence and distribution of specific compound classes within the full collection. A cluster was identified that was primarily related to the *Pantoea* genus. Together with the results from biological screening of the 53 strains collection, we identified a series of structurally related compounds that could be responsible for the observed larvicidal activity. This was confirmed by injecting the isolated compounds **1**–**3**, which were effectively found to be present in this particular cluster. Interestingly, this cluster of lipoamino acid-related structures indicated the following: (i) There is high structural diversity in the compounds generated by the *Pantoea* genus, and (ii) some of the members of this structural family were also found to be present in other entomopathogenic strains of the collection (Figure 3).

To study in more depth the chemistry of *Pantoea* lipoamino acids, a molecular network was generated organizing the tandem MS data acquired from the *Pantoea* sp. extracts together with the MS data generated from the isolated and synthetic lipoamino acid derivatives (Figure 4). In addition to the three isolated lipoamino acids (**1**–**3**), it was possible to align six of the synthetic lipoamino acids (**4**, **8**, **9**, **14**, **20** and **22**) with features present in the crude *Pantoea* sp. extracts, thus indicating that these compounds were also naturally present in the bacterium extract (represented as squares in Figure 4). Two of the synthesized derivatives (**5** and **10**) were also found to be present in the cluster but were not detected in the natural extracts (arrow-like node in Figure 4).

The co-injection of synthetic analogues indicated that the studied extract contained additional lipoamino acid derivatives and allowed us to propagate annotations to neighboring nodes. To further annotate such compounds in the cluster, the mass spectral signature of each individual node was manually curated (Figure 5). For each feature, the molecular formula was established, annotations were made regarding adducts or complexes, and each fragmentation spectrum was treated using Sirius GUI software [17]. The results are summarized in Appendix A and allowed us to putatively annotate 16 additional compounds. Inspection of the fragmentation pattern revealed that five diagnostic fragments corresponding to the amino acid portion of the compounds could be observed within this cluster. The phenylalanine (Phe) and Phe-methyl ester [M+H]^+^ moieties gave a typical fragment at *m*/*z* 166.09 and 188.10, respectively. Fragments at *m*/*z* 132 and 146 indicated potential leucine/isoleucine (Leu/Ile) and Leu/Ile-methyl ester [M+H]^+^ moieties, respectively. Finally, a fragment at *m*/*z* 118 indicated a potential valine (Val) [M+H]^+^ moiety. It is to be noted that an additional lipoamino acid derivative bearing a tyrosine-methyl ester could also be identified in another cluster of the molecular network by co-injection with a synthetic analogue (**19**). The presence of double bonds was inferred from the calculated round double bond equivalent (RDBE). The position of the OH groups was inferred from the isolated compounds. Together with the molecular formula determination, these signature indications allowed us to propose putative partial structures for the other nodes of the cluster and highlights the significant diversity of the lipoamino acids produced by this *Pantoea* sp. strain (Figure 5).

## 3. Materials and Methods

### 3.1. General Procedures

Optical rotations were obtained on an Anton Paar MCP 200 polarimeter (Anton Paar Graz, Austria) in a 100 mm-long 350 µL cell using MeOH as the solvent at 20 °C. High resolution ESITOFMS measurements were performed using a Waters ACQUITY UHPLC system coupled to a Waters Micromass LCT-Premier Time-of-Flight mass spectrometer equipped with electrospray interface (ESI) (Waters, Manchester, England). Nuclear magnetic resonance (NMR) data were recorded in CD_3_OD on Bruker 500 MHz and 600 MHz spectrometers equipped with a 1 mm inverse detection probe. Chemical shifts (*δ*) are reported in ppm based on the TMS signal, and coupling constant (*J*) is reported in Hertz (Bruker, Rheinstetten, Germany). Flash chromatography was performed on a Grace Reveleris system with UV and ELSD detectors using 120 g C18 or 80 g silica gel columns (Grace, Maryland, USA). HPLC analysis was conducted on a Gilson system equipped with a 322 pumping device, a GX-271 fraction collector, a 171 diode array detector, and a prepELSII detector electrospray nebulizer (Gilson Middelton, USA). Analytical analysis was conducted using Phenomenex Luna C18 (5 µm, 4.6 × 250 mm) and Phenomenex Kinetex C8 (5 µm, 4.6 × 250 mm) columns. A preparation analysis was conducted using Phenomenex Luna C18 (5 µm, 21.2 × 250 mm) and Kinetex C8 (5 µm, 21.2 × 250 mm) columns (Phenomenex, Le Pecq, France). All the chemicals were purchased from Sigma-Aldrich (Sigma-Aldrich chimie, Saint-Quentin-Fallavier, France).

### 3.2. Collection and Identification of Pantoea sp. SNB-VECD14B

An individual insect from the Diaspididae family that was infected by entomopathogenic microorganisms was collected in Montsinéry, French Guiana. The cuticle of the insect was scraped with a handle and transplanted onto a Petri dish containing a solid potato dextrose agar (PDA) medium and then stored at 28 °C. After one day, growing bacteria were removed and transferred onto another Petri dish. The strain SNB-VECD14B was stored in triplicate at −80 °C in a H_2_O–glycerol mixture (50:50). A sample was submitted for amplification of the nuclear ribosomal internal transcribed spacer region 16S, which allowed for identification after comparison to the NCBI sequence. The sequence was registered in the NCBI GenBank database with the accession number KX858894 and identified as *Pantoea* sp. A molecular analysis was performed externally by BACTUP, France.

### 3.3. Culture, Extraction, and Isolation

The bacteria strain was initially cultivated on a small scale using Petri dishes with Mueller Hinton (MH) solid medium and then on a large scale using 176 14 cm Petri dishes at 28 °C for 8 days. The culture medium containing the mycelium was cut into small pieces and macerated with ethyl acetate (EtOAc) at room temperature on a rotary shaker (70 rpm) for 48 h. The contents were extracted three times with 5 L of EtOAc using a separatory funnel. The combined organic layers were washed with water. The organic solvent was evaporated to dryness under reduced pressure to yield the crude extract (2.5 g). A portion of the crude extract (2.35 g) was fractionated by reversed-phase flash chromatography on a C18 column using a linear gradient of water—acetonitrile (*v*/*v*, 1:0 to 0:1 over 20 min, flow rate = 80 mL/min) followed by another gradient of acetonitrile—methylene chloride (v/v, 1:1 to 0:1 over 10 min, flow rate = 80 mL/min) to generate 8 fractions that were labeled A to H. The larvicidal and antimicrobial activities were concentrated in fractions E (60.5 mg), F (445.7 mg), G (898.7 mg) and H (108.4 mg). 

Fraction E was fractionated by preparative HPLC using H_2_O–ACN (37:63 to 26:74 over 30 min, flow rate = 21 mL/min) to afford compound **1** (1.9 mg, t_R_ = 20.40 min) and compound **2** (1.2 mg, t_R_ = 24.10 min).

Fraction F was fractionated by preparative HPLC using H_2_O–ACN (33:67 to 17:83 over 30 min, flow rate = 21 mL/min) to afford compound **3** (10.3 mg, t_R_ = 17.10 min).

Compound **1**: White powder; ${\left[\alpha \right]}_{D}^{20}$+ 29 (*c* 0.2, MeOH); ^1^H and ^13^C NMR spectroscopic data, see Appendix A; HRESITOFMS *m/z* 392.2769 [M+H]^+^ (calculated for C_23_H_38_NO_4_^+^, 392.2795). MSMS spectra were deposited at CCMSLIB00004684200.

Compound **2**: White powder; ${\left[\alpha \right]}_{D}^{20}$ + 6 (*c* 0.2, MeOH); ^1^H and ^13^C NMR spectroscopic data, see Appendix A; HRESITOFMS *m/z* 418.2926 [M+H]^+^ (calculated for C_25_H_40_NO_4_^+^, 418.2952). MSMS spectra were deposited at CCMSLIB00004684202.

Compound **3**: White powder; ${\left[\alpha \right]}_{D}^{20}$+ 12 (*c* 0.2, MeOH); ^1^H and ^13^C NMR spectroscopic data, see Appendix A; HRESITOFMS *m/z* 402.3004 [M+H]^+^ (calculated for C_25_H_40_NO_3_^+^, 402.3003). MSMS spectra were deposited at CCMSLIB00004684204.

### 3.4. General Synthetic Procedure for Lipoamino Acid Methyl Ester Synthesis

To a mixture of the fatty acid (100 mg), amino acid methyl ester (1 equiv.) and HATU (1.5 equiv.) was added Dimethylformamide (DMF) (5 mL) followed by *N*,*N*-Diisopropylethylamine (DIEA) (3 equiv.) under a N_2_ atmosphere. The reaction mixture was stirred at room temperature (RT) overnight. A saturated solution of ammonium chloride (NH_4_Cl) was added and the reaction mixture was extracted with *tert*-butylmethylether (3 times). The combined organic phases were dried with Na_2_SO_4_ and concentrated in vacuo. The crude mixture was purified by flash chromatography on silica gel to provide compounds **4**–**21**. Analytical data including optical rotations are reported in the Appendix A.

### 3.5. General Synthetic Procedure for Lipoamino Acid Synthesis

To a solution of the lipoamino acid methyl ester (50 mg) in methanol (5 mL) and water (0.5 mL) was added LiOH.H_2_O (5 equiv.). The reaction mixture was stirred at RT for 4 h and then aqueous HCl (1 M) was added to adjust the pH to 2–3. The resulting mixture was extracted twice with ethyl acetate, dried with Na_2_SO_4_, and concentrated in vacuo to provide the pure compounds **22**–**39**. Analytical data, including optical rotations, are reported in the Appendix A.

### 3.6. Synthesis of Compound **3**

To a mixture of the palmitoleic acid (50 mg), L-phenylalanine methyl ester (80 mg) and HATU (143 mg) was added DMF (5 mL) followed by DIEA (97 mg) under a N_2_ atmosphere. The reaction mixture was stirred at RT overnight. A saturated solution of ammonium chloride (NH_4_Cl) was added and the reaction mixture was extracted with *tert*-butylmethylether (3 times). The combined organic phases were dried with Na_2_SO_4_ and concentrated in vacuo. The crude mixture was purified by flash chromatography using C18 4g silica gel to give 52.5 mg of **3-OMe**. 

To a solution of **3-OMe** (15 mg) in MeOH (5ml) and water (0.5 mL) was added LiOH.H_2_O (5 equiv.). The reaction mixture was stirred at RT for 4 h and aqueous HCl (1M) was added to adjust the pH to 2–3. The resulting mixture was extracted twice with ethyl acetate, dried with Na_2_SO_4_ and concentrated in vacuo to provide 14 mg of **3**. ${\left[\alpha \right]}_{D}^{20}$ +49 (*c* 0.5, CHCl_3_); HRESITOFMS *m/z* 402.2976 [M+H]^+^ (calculated for C_25_H_40_NO_3_^+^, 402.3003); ^1^H and ^13^C NMR spectroscopic data, see S76-S77.

The same synthesis was done with D-phenylalanine methyl ester to give ***ent-*3-OMe** (65.6 mg) and ***ent*-3** (13.5 mg) with ${\left[\alpha \right]}_{D}^{20}$-49 (*c* 0.5, CHCl_3_); HRESITOFMS *m/z* 402.2977 [M+H]^+^ (calculated for C_25_H_40_NO_3_^+^, 402.3003); ^1^H and ^13^C NMR spectroscopic data, see S100-S101.

### 3.7. Molecular Networking

#### 3.7.1. Mass Spectrometry Analysis Parameters

Chromatographic separation was performed on a Waters ACQUITY UPLC system interfaced to a Q-Exactive Focus mass spectrometer (Thermo Scientific, Bremen, Germany), with a heated electrospray ionization (HESI-II) source. Thermo Scientific Xcalibur 3.1 software was used for instrument control. The LC (Liquid Chromatography) conditions were as follows: Column, Waters BEH C_18_ 50 × 2.1 mm, 1.7 μm; mobile phase, (A) water with 0.1% formic acid, (B) acetonitrile with 0.1% formic acid; flow rate, 600 μL·min^−1^; injection volume, 1 μL; gradient, linear gradient of 2−100% B over 6 min and then isocratic at 100% B for 0.6 min. An ACQUITY UPLC photodiode array detector was used to acquire PDA spectra, which were collected from 210 to 450 nm. In positive ion mode, the diisooctyl phthalate C_24_H_38_O_4_ [M+H]^+^ ion (*m/z* 391.28429) was used as the internal lock mass. The optimized HESI-II parameters were as follows: Source voltage, 4.0 kV (pos); sheath gas flow rate (N_2_), 55 units; auxiliary gas flow rate, 15 units; spare gas flow rate, 3.0; capillary temperature, 275.00 °C (pos), S-lens RF level, 45. The mass analyzer was calibrated using a mixture of caffeine, methionine−arginine−phenylalanine−alanine−acetate (MRFA), sodium dodecyl sulfate, sodium taurocholate, and Ultramark 1621 in an acetonitrile/methanol/water solution containing 1% formic acid by direct injection. The data-dependent MS/MS events were performed on the three most intense ions detected in the full scan MS (Top3 experiment). The MS/MS isolation window width was 1 Da, and the stepped normalized collision energy (NCE) was set to 15, 30, and 45 units. In data-dependent MS/MS experiments, full scans were acquired at a resolution of 35,000 FWHM (at *m/z* 200) and MS/MS scans at 17,500 FWHM both with an automatically determined maximum injection time. After being acquired in a MS/MS scan, parent ions were placed in a dynamic exclusion list for 2.0 s.

#### 3.7.2. MS Data Pretreatment

The MS data were converted from the .RAW (Thermo) data format to the .mzXML format using MSConvert software, which is part of the ProteoWizard package [15]. The converted files were treated using the MzMine software suite [17]. 

The parameters were adjusted as follows: The centroid mass detector was used for mass detection with the noise level set to 1.0 × 10^6^ when the MS level was set to 1, and to 0 when the MS level was set to 2. The ADAP chromatogram builder was used and set to a minimum group size of 5 scans, minimum group intensity threshold of 1.0 × 10^5^, minimum highest intensity of 1.0 × 10^5^, and *m/z* tolerance of 8.0 ppm. For chromatogram deconvolution, the wavelets (ADAP) algorithm was used. The intensity window S/N was used as the S/N estimator with a signal-to-noise ratio set at 25, a minimum feature height at 10,000, a coefficient area threshold at 100, a peak duration range from 0.02 to 0.9 min and the RT wavelet range from 0.02 to 0.05 min. Isotopes were detected using the isotope peak grouper with a m/*z* tolerance of 5.0 ppm, an RT tolerance of 0.02 min (absolute), and the maximum charge set at 2. The representative isotope used was the isotope that was the most intense. An adduct (Na^+^, K^+^, NH_4_^+^, ACN^+^, CH_3_OH^+^, Isoprop^+^) search was performed with the RT tolerance set at 0.1 min and the maximum relative peak height at 500 %. A complex search was also performed using [M+H]^+^ in ESI positive mode, with the RT tolerance set at 0.1 min and the maximum relative peak height at 500%. Finally, a custom database search was performed using the Dictionary of Natural Products 2018 (v. 26.2) database (http://dnp.chemnetbase.com) where the search was restricted to fungal or bacterial metabolites. Peak alignment was performed using the join aligner method where the *m*/*z* tolerance was set at 8 ppm, absolute RT tolerance at 0.065 min, weight for *m/z* at 10 and weight for RT at 10. The peak list was gap-filled with the same RT and *m/z* range gap filler (*m*/*z* tolerance at 8 ppm). Eventually the resulting aligned peak list was filtered using the peak-list rows filter option to keep only those features associated with MS2 scans. Full parameters are available as a .xml file in the Appendix A (Entomopathogens_MzMine_parameters.mzmine).

#### 3.7.3. Molecular Networks Generation

To maintain the retention time and exact mass information and to allow the separation of isomers, the molecular networks were created using the .mgf file resulting from the MzMine pretreatment step detailed above: https://bix-lab.ucsd.edu/display/Public/Feature+Based+Molecular+Networking. Spectral data were uploaded on the Global Natural Products Social molecular networking platform [10]. A network was then created where edges were filtered to have a cosine score above 0.65 and more than 6 matched peaks. Further edges between two nodes were kept in the network if and only if each of the nodes appeared in each other’s respective top 10 most similar nodes. The spectra in the network were then searched against GNPS spectral libraries. The library spectra were filtered in the same manner as the input data. All matches kept between network spectra and library spectra were required to have a score above 0.7 and at least 6 matched peaks. The output was visualized using Cytoscape 3.6 software [18]. The full MS data set was uploaded and is accessible on the GNPS servers as Massive Data sets N° MSV000082940. The molecular network analysis and clustered spectra are accessible at the following link: https://gnps.ucsd.edu/ProteoSAFe/status.jsp?task=b036dd9fcb964ca49fe8dc4345944b86.

### 3.8. Evaluation of Larvicidal Activity

#### 3.8.1. Insect Collection and Rearing

The *Aedes aegypti* strain was used for extract and compound Testing. The laboratory strain Paea, came from French Polynesia and had been maintained at the Institut Pasteur de la Guyane in French Guiana for over a decade. This strain is susceptible to all insecticides. *Ae. aegypti* eggs were conserved on dried paper strips. Hatching was induced by dropping these strips in water and placing them under vacuum pressure for at least 1 h. The larvae were fed with yeast pellets. Larval rearing was performed under natural conditions at 28 °C ± 2 °C, 80 % ± 20 % RH and with a photoperiod of 14 h dark and 10 h light. Late third or early fourth-instar larvae were used in the tests. All crude extracts and fractions were investigated using the WHO procedure for testing mosquito larvicides [19,20]. The larvicidal activity of pure compounds was evaluated using a tube assay.

#### 3.8.2. Cup Assays

A total of 100 larvae were exposed in each bioassay. Four replicates of 25 larvae were prepared in cups containing 99 mL of distillated water. A 1 mL aliquot of the extract/fraction in absolute ethanol was added to each cup. The crude extract and the fractions were all tested at a concentration of 100 ppm. Absolute ethanol (1 mL) served as a negative control. Larval mortality was recorded 24 h after exposure.

#### 3.8.3. Tube Assays

Fifty larvae were tested at each concentration with ten replicates per concentration (10 test tubes × 5 larvae). Each compound was tested at a concentration of 10 μg/mL. A stock solution was prepared at 1 mg/mL in absolute ethanol, and then, 30 μL of this solution was added to each test tube (75 × 12 × 0.8 − 1.0 soda rimLess, catalog #212-0013, VWR, International) containing 2.97 mL of distilled water. Larval mortality was recorded 24 h after exposure.

#### 3.8.4. Statistical Analysis

Abbott’s formula for mortality was applied if the mortality rate in the control was between 5 and 20%. The test was invalidated if the mortality rate in the control was greater than 20% [21].

### 3.9. Antimicrobial Assays

The ATCC strains were purchased from the Pasteur Institute and clinical isolates were provided by Phillipe Loiseau (University Paris Sud, Châtenay-Malabry, France). The strains used in this study were: *Candida albicans* ATCC10231, *Staphylococcus aureus* ATCC29213, methicillin-resistant *S. aureus* ATCC33591, and *Trichophyton rubrum* SNB-TR1. Extracts, fractions, and pure compounds were tested according to the reference protocol of the European Committee on Antimicrobial Susceptibility Testing (EUCAST, http://www.eucast.org, 11 April 2016). The standard microdilution test as described by the Clinical and Laboratory Standards Institute guidelines (M27-A2, M7-A8 and M38-A) was used to determine minimal inhibition concentrations (MIC) against dermatophyte fungi, bacteria, and yeasts [22,23,24]. Crude extracts and pure compounds were tested at concentrations ranging from 256 to 0.5 μg/mL. The microplates were incubated at 32 °C, and MIC values were calculated after 24 h for bacteria, 48 h for yeast and 5 days for *T. rubrum*. The MIC values reported in Table 2 refer to the lowest concentration preventing visible growth in the wells. All assays were conducted in triplicate. 

## 4. Conclusions

In conclusion, our study highlights that *Pantoea* sp. SNB-VECD14B can biosynthesize diverse lipoamino acids with antimicrobial and/or insecticidal activities. Three hydroxyacyl-phenylalanine derivatives were isolated and characterized from EtOAc extracts, and different analogs were synthesized to demonstrate structure-activity relationships. In short, we observed that the free carboxylic acid group was important for biological activity. Compounds with a long carbon chain (16 to 18 carbons) and a *Z* double bond, exhibited the highest larvicidal activity, while compounds with a saturated C12 or C14 chain demonstrated the best antimicrobial activity. The studied lipoamino acids appear to have similar biological and structural properties compared to free fatty acids [25,26]. Their amphipathic nature provides a wide range of activities. Given the observed biological activities, these molecules could provide an alternative to common antimicrobial agents for application in agriculture, food preservation, or cosmetics. 

Finally, the molecular networking-based approach revealed several close analogues of lipoamino acids in the *Pantoea* sp. extract. The global molecular network established over a wide collection of entomopathogenic strains indicated that this structural family was indeed typical of the *Pantoea* genus. These lipoamino acids have diverse biological activities, and it can be hypothesized that the strain has acquired the ability to protect itself from various environmental aggressions (fungi, bacteria, and insects) by producing a truly diverse mix of lipoamino acids that exhibit complementary activities.

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
