# Peer review of "Characterization, Diversity, and Structure-Activity Relationship Study of Lipoamino Acids from Pantoea sp. and Synthetic Analogues"

_ijms, 2019, doi:10.3390/ijms20051083_

Reviewer 1 Report

The authors report on natural and synthetic lipo-amino acids and their antimicrobial and insecticidal activities.

The MIC values for pure sample must be given also (or only)  in µMolar, so that it is possible to compare the data with compounds taken as positive control ( data to be introduced) , eg. In table 2 , as in other part of the manuscript, abstract included for compound 3 . Herein, the MIC value of 16 µg/mL is given for metabolite 3 and the same value is recorded for the extract at line 21, page 2 : this must be checked, or an explanation must be inserted.

In figure 1, the stereochemistry at C-2 is not clear:  both the full and empty wedge must be indicated.

The assignment of the configuration at C-2 is not clearly illustrated in the text: in figure 1 seems to want to be indicated, but then at line 8, page 3 the nomenclature does not describe it ;  an indication that it will be defined below by synthesis must be introduced at this point of the discussion. This is also the case of the position and configuration of the unsaturation in the chain: the authors must discuss better the citation at reference 13 , or refer to the synthesis discussed below. L and D notation are appropriate for the kind of amino acids used in the synthesis, but the configuration at C-2 is more properly defined as R/S.

At page 4, line 17, the assumption:  The configuration of C-3’ could not be determined. However, in view of the specific optical rotation observed, we can assume that these…” is not rigorous because in compound 3 another stereogenic centre is present  and the comparison of optical activity is not completely correct; the data for the presence of pure and not mixture of diasteroisomers could come by an accurate LC-MS analysis.

At page 5, line 8, the authors must introduce a reference for the synthetic method used.

At page 8, line 17: specify the meaning of stereochemistry, if E/Z configuration and/or  R/S, or both.

At page 11, line 37: indicate the supplier company for  the fatty acids and other reagents.

 In Supplementary, the optical activity values for all the synthetic products must be given.

Minor comments:

Caption to Figure 1: “Pantoea” in italic form

Caption to Scheme 1: compounds numbering in bold

Page 12. Line 10 : “in vacuo” in italic

Page 15, line 21, remove “of” from the abbreviated title of the journal

Page 16, line 8, replace BMC Bioinformatics to BMC Bioinf.

Author Response

Dear reviewer,

major corrections have been done thanks to your comments (they are indicated in blue in the manuscript) and explications are described below.

The MIC values for pure sample must be given also (or only)  in µMolar, so that it is possible to compare the data with compounds taken as positive control ( data to be introduced) , eg. In table 2 , as in other part of the manuscript, abstract included for compound 3 .

For MIC values, CLSI standard protocoles are used (ref 22-24 included in the manuscript). These documents are « reference » standard developed through a consensus process to facilitate agreement among laboratories in measuring the susceptibility of antifungal and antibacterial agents. It is specified that MICs should be given in µg/mL. The positive controls in table 2 are in µg/mL as well.

Herein, the MIC value of 16 µg/mL is given for metabolite 3 and the same value is recorded for the extract at line 21, page 2 : this must be checked, or an explanation must be inserted.

The AcOEt extract of Pantoa sp. showed a MIC value of 16 µg/mL against T. rubrum. No isolated compound gave the same MIC on this pathogenic fungus. Active compounds may be in small amount in the extract. Our group and others have also demonstrated that synergies can sometimes account for the antimicrobial potential of some extracts.

In figure 1, the stereochemistry at C-2 is not clear:  both the full and empty wedge must be indicated. The assignment of the configuration at C-2 is not clearly illustrated in the text: in figure 1 seems to want to be indicated, but then at line 8, page 3 the nomenclature does not describe it ;  an indication that it will be defined below by synthesis must be introduced at this point of the discussion. This is also the case of the position and configuration of the unsaturation in the chain: the authors must discuss better the citation at reference 13 , or refer to the synthesis discussed below. L and D notation are appropriate for the kind of amino acids used in the synthesis, but the configuration at C-2 is more properly defined as R/S.

Thank you for this comment. Yes, the stereochemistry has been established by synthesis and it is best to make this clear in the structural elucidation section. All the details have been added to the document, including an explanation on the stereochemistry of the double bond. Just a small point regarding the stereochemistry at C-2 in Figure 1: Several journals would generally require one and only one stereochemical indication per atom. However we do not know what in preferred at the International Journal of Molecular Science. Please let us know how the journal wants these structures drawn. Finally, lipoaminoacids have been referred as (S) or (R) rather than L or D.

At page 4, line 17, the assumption:  The configuration of C-3’ could not be determined. However, in view of the specific optical rotation observed, we can assume that these…” is not rigorous because in compound 3 another stereogenic centre is present  and the comparison of optical activity is not completely correct; the data for the presence of pure and not mixture of diasteroisomers could come by an accurate LC-MS analysis.

Indeed it is an error. Sorry about this, the sentence has been removed

At page 5, line 8, the authors must introduce a reference for the synthetic method used.

Done, see reference number 16

At page 8, line 17: specify the meaning of stereochemistry, if E/Z configuration and/or  R/S, or both.

Some correction has been done in this paragraph. You’re right, the sentence was very confusing.

At page 11, line 37: indicate the supplier company for the fatty acids and other reagents.

It as been implemented in general procedure.

In Supplementary, the optical activity values for all the synthetic products must be given.

Done see Table S5 in supporting information

Minor comments:

Caption to Figure 1: “Pantoea” in italic form

Caption to Scheme 1: compounds numbering in bold

Page 12. Line 10 : “in vacuo” in italic

Page 15, line 21, remove “of” from the abbreviated title of the journal

Page 16, line 8, replace BMC Bioinformatics to BMC Bioinf.

All errors were corrected

Reviewer 2 Report

Dear Authors,

The manuscript ID: ijms-447103-v1 entitled “Characterization, diversity, and structure-activity relationship study of lipoamino acids from Pantoea sp. and synthetic analogues” written by Seindé Touré, Sandy Desrat, Léonie Pellissier, Pierre-Marie Allard, Jean-Luc Wolfender, Isabelle Dusfour, Didier Stien and Véronique Eparvier is very interesting.

This article contains a fair bit of important information about the lipoamino acids from Pantoea sp. and their synthetic derivatives. The structures of these compounds were explained using NMR and MS spectroscopic data analysis. The antimicrobial and insecticidal activity were also showed. Moreover, using a molecular networking-based approach, several analogues of the newly synthesized lipoamino acids were discovered in the Pantoea sp. extract, which was found to be very insightful. These studies are well documented and presented.

However, I think that the manuscript has some small drawbacks that must be addressed before publication. 

1.    Why only such microorganisms were used for research of antimicrobial activity? As the authors noted, many of the pathogens are becoming resistant to nearly all of the antibiotic drugs currently available and are major clinical problem. Therefore, you may consider including some other relevant resistant gram-positive or negative bacteria in your study, as that would greatly increase the significance of the paper.

2.    Please provide more details about the antimicrobial assay (Page 14). What concentration range was used in the study?

3.    Please add reference numbers of microorganisms in Table 2.

I have no other comments on the manuscript.

Page 2; L. 29: investigation – investigation.

Page 2; L. 32: Pantoea – Pantoea (italic)

Page 15; L 38: Curr. Opin. Chem. Bio – Curr. Opin. Chem. Biol.

Best regards,

Author Response

Dear reviewer, thank you for yoyr comments, we corrected manuscrit  thanks to you remarks,

1.    Why only such microorganisms were used for research of antimicrobial activity? As the authors noted, many of the pathogens are becoming resistant to nearly all of the antibiotic drugs currently available and are major clinical problem. Therefore, you may consider including some other relevant resistant gram-positive or negative bacteria in your study, as that would greatly increase the significance of the paper.

During this project we wanted to highlight both insecticidal and antimicrobial activities in our extract strains. We chose in this study to test fungi, bacteria and yeasts. The isolated quantities of pure molecules from the resource are low which did not allow us to test on other resistant strains. This can be done in a future project.

2.    Please provide more details about the antimicrobial assay (Page 14). What concentration range was used in the study?

Details have been included in Experimental section.

3.    Please add reference numbers of microorganisms in Table 2.

 Done

I have no other comments on the manuscript. 

Page 2; L. 29: investigation – investigation.

Page 2; L. 32: Pantoea – Pantoea (italic)

Page 15; L 38: Curr. Opin. Chem. Bio – Curr. Opin. Chem. Biol.

Any errors or omissions found during the verification were corrected

Round  2

Reviewer 1 Report

The authors have considered all comments, with detailed answers and modification of the manuscript.

At page 2, line 34, Pantoea in italic form.

Reviewer 2 Report

Dear Authors,

The manuscript ID: ijms-447103-v2 entitled “Characterization, diversity, and structure-activity relationship study of lipoamino acids from Pantoea sp. and synthetic analogues” written by Seindé Touré, Sandy Desrat, Léonie Pellissier, Pierre-Marie Allard, Jean-Luc Wolfender, Isabelle Dusfour, Didier Stien and Véronique Eparvier was corrected according to review’s suggestions. I have no major comments on this article.

Best regards,